# Recombinant Forms of HIV-1 in the Last Decade of the Epidemic in the Russian Federation

**DOI:** 10.3390/v15122312

**Published:** 2023-11-25

**Authors:** Anastasiia Antonova, Elena Kazennova, Aleksey Lebedev, Ekaterina Ozhmegova, Anna Kuznetsova, Aleksandr Tumanov, Marina Bobkova

**Affiliations:** 1The National Research Center for Epidemiology and Microbiology Named after Honorary Academician N.F. Gamaleya of the Ministry of Health of the Russian Federation, 123098 Moscow, Russia; kazennova@rambler.ru (E.K.); lebedevalesha236@gmail.com (A.L.); belokopytova.01@mail.ru (E.O.); a-myznikova@list.ru (A.K.); desep@mail.ru (A.T.); 2I. Mechnikov Research Institute for Vaccines and Sera, 105064 Moscow, Russia; mrbobkova@mail.ru

**Keywords:** HIV-1, subtyping, phylogenetic analysis, CRFs, URFs, Russia, drug resistance mutations

## Abstract

Currently, HIV-1 displays a substantial level of genetic diversity on a global scale, partly attributed to its recombinant variants. This study seeks to identify and analyze HIV-1 recombinants in Russia during the last decade of the epidemic. A comprehensive examination was conducted, encompassing 3178 partial pol sequences. Subtyping was achieved through various programs including COMET, the Stanford Database, REGA, jpHMM, RIP, and RDP4 for recombination analysis. The study also involved phylogenetic analysis to trace the origins of the identified recombinants. Primary resistance (PrimDR) prevalence and Drug Resistance Mutations (DRMs) were assessed. The study uncovered an overall proportion of recombinants at 8.7%, with a statistically significant increase in their frequency observed over time (*p* < 0.001). The Northwestern (18.5%) and Siberian (15.0%) Federal Districts exhibited a high prevalence of recombinants, while the Volga (1.9%) and Ural (2.8%) Federal Districts had a lower prevalence. Among HIV-1 recombinants, a PrimDR prevalence of 11.4% was identified. Notably, significant differences in DRMs were observed, with a higher prevalence of M184V in sub-subtype A6 (*p* = 0.018) and K103N in CRF63_02A6 (*p* = 0.002). These findings underscore the increasing HIV-1 genetic diversity and highlight a substantial prevalence of PrimDR among its recombinant forms, emphasizing the necessity for ongoing systematic monitoring.

## 1. Introduction

Currently, the HIV/AIDS epidemic remains a global public health issue worldwide. According to the Joint United Nations Program on HIV/AIDS (UNAIDS), the global number of people living with HIV (PLWH) was 39.0 million [33.1–45.7 million] people in 2022 [1]. In the Russian Federation, this figure exceeded 1 million people [2].

According to regional data, Eastern and Southern Africa remains the most affected region in the world, with 20.8 million [17.4–24.5 million] PLWH in 2022, representing 53.3% of the global number of PLWH. The proportion of AIDS-related deaths in this region was 41.2% of the global rate [3]. In other major regions of the world, such as Western and Central Europe and North America and Eastern Europe and Central Asia (as classified by UNAIDS), the number of PLWHA was 2.3 million [1.9–2.6 million] and 2.0 million [1.8–2.1 million], respectively. However, the Eastern Europe and Central Asia region leads in the number of new cases of HIV infections, with 160,000 [140,000–180,000] people in 2022 [3]. And the Russian Federation has the highest rates of new cases of HIV infections in this region [4].

One of the components of the HIV surveillance package is molecular genetic monitoring, which makes it possible to identify HIV genetic variants circulating in a particular territory or within risk groups. The HIV pandemic is based on the global circulation of HIV-1 group M, which includes 10 independent subtypes: A–D, F–H, J–L, and circulating recombinant forms (CRFs) of the virus [5]. Together, they account for more than 90% of all HIV infections worldwide. Some HIV-1 subtypes are additionally subdivided into sub-subtypes: A1–A8 and F1–F2; their nucleotide sequences differ by no more than 20%, but form separate branches on phylogenetic trees [6].

The distribution of HIV-1 genetic variants is diverse both globally and within individual regions and countries [7]. In the early years of the HIV epidemic in Russia, subtype B dominated the genetic structure, detected in 99% of men who have sex with men (MSM) and in 23% of heterosexual infections. Additionally, approximately 29% of patients registered before 1996 were infected with subtype G, linked to a nosocomial outbreak in Elista in 1988 [8].

Starting from 1995, a rapid shift occurred in the HIV-1 genetic landscape in Russia. A distinctive epidemic pattern emerged, with the parenteral route through drug injection becoming the primary mode of transmission, accounting for roughly 90% of all recorded cases [9]. Concurrently, the sub-subtype A6 virus (IDU-A or A-FSU) infiltrated the community of injecting drug users (IDUs), swiftly spreading across all regions of the country with minimal genetic alterations. This led to a prolonged period of genetic uniformity within HIV-1 in Russia [10,11].

Currently, there is a notable global increase in HIV-1 genetic diversity, attributed in part to its recombinant forms, rising from 9.3% in 1990–1999 to 22.8% in 2010–2015 [12]. A molecular epidemiological analysis of HIV-1 variants in Russia from 1987–2015 reveals a 20% reduction in the prevalence of subtype A1 (specifically sub-subtype A6), accompanied by a rise in other viral genetic variants, including recombinant forms [11]. A systematic review and meta-analysis indicate that by mid-2017, HIV-1 recombinant forms accounted for 21.3% in Russia and former Soviet Union countries, displaying substantial heterogeneity [13]. Within the Russian Federation, two circulating recombinant forms of HIV-1 have been identified directly:CRF03_AB—in the late 1990s in Kaliningrad—is the result of recombination between viruses of genetic variants A6 and B;CRF63_02A6—in 2009–2010 in Siberia—is a secondary recombinant formed by CRF02_AG and sub-subtype A6 [14,15].

A number of studies have noted the impact of recombination on the biological properties (high pathogenicity) of HIV-1, as well as on the efficacy of current antiretroviral therapy (ART) due to its the accelerating effect on the formation of HIV-1 drug resistance (HIV DR) mutations [16,17,18,19,20,21]. Research involving the modeling of HIV-1 evolution dynamics in response to host immune system pressure has indicated that recombination can drive the rapid adaptation and survival of viruses reactivated from latent reservoirs [22].

Therefore, the increasing prevalence of HIV-1 recombinant variants may play an important role in the development and spread of pretreatment drug resistance (PDR), which complicates the prevention and treatment of the HIV infection and reduces the effectiveness of ART.

The epidemic process of the HIV infection in Russia is an area of great interest. Russia has a large territory inhabited by people of different age categories and nationalities, with differences in customs and behavior, lifestyle, and mobility patterns. The Russian Federation borders on 18 states, including eight FSU countries. All this contributes to the high HIV-1 genetic diversity in Russia, and leads to the formation and spread of its recombinant forms.

The aim of our study was to identify and analyze the HIV-1 recombinant forms in Russia in the last decade of the epidemic (from 2011 to 2020).

## 2. Materials and Methods

### 2.1. Study Population and Data Collection

The material of the study were blood plasma collections (*n* = 2502), peripheral blood mononuclear cells (lymphocytes) collections (*n* = 613), and frozen whole blood collections (*n* = 63) obtained from HIV-infected patients at federal and regional centers for HIV/AIDS prevention (AIDS Centers) as part of molecular genetic monitoring of the HIV epidemic in Russia in the period from 2011 to 2020. We analyzed sequences and their associated demographic and epidemiological data. The sequences covered the HIV-1 protease (PR) and part of the reverse transcriptase (RT).

Informed consent was signed by all HIV-infected patients included in the study in accordance with the Declaration of Helsinki. All the data were anonymized and coded.

### 2.2. RNA and DNA Extraction, Amplification, and HIV-1 Sequencing

Blood collection and fractionation procedures to obtain lymphocytes and blood plasma were carried out by employees of federal and regional AIDS Centers.

The ViroSeqTM HIV-1 Genotyping System kit (Celera Diagnostics, Alameda, CA, USA) was used for RNA extraction from blood plasma samples. Commercial kit (the QIAamp DNA Blood Mini Kit (“QIAGEN”, Hilden, Germany)) was used for DNA extraction from lymphocytes and frozen whole blood.

Amplification of HIV-1 *pol* gene fragments was performed automatically by nested polymerase chain reaction (PCR) method according to the protocols presented in Table 1.

PCR conditions are presented in Appendix A.

In case of working with HIV-1 RNA, the first step involved performing a reverse transcription reaction as described in the ViroSeq HIV-1 Genotyping System user manual [23].

To purify amplified DNA fragments from the reaction mixture, a commercial kit using magnetic particle suspension (“Sileks”, Moscow, Russia) and a commercial Cleanup S-Cap kit (“Evrogen”, Moscow, Russia) were used.

Sanger-based sequencing of the HIV *pol* gene regions encoding the PR-RT (2253–3554 bp according to the HXB-2 strain, GenBank accession number K03455) was performed. Sequencing reaction for amplification protocols A and B (Table 1) was performed using BigDye Terminator v3.1 Cycle Sequencing Kit (“Applied Biosystems“, Waltham, MT, USA) according to the manufacturer’s protocol. The set of primers depended on amplification protocol (Table 1):for A protocol: PRsec1 (CAAAAATTGGGCCTGAAAATCCATA), PPOS2 (GCTAATTTTTTAGGGAAGATCTGGCCTT), RTsec1 (CAAAAATTGGGCCTGAAAATCCATA), RTOA (TGCCTCTGTTAATTGTTTTACATCATTAGTGTG);for B protocol: F2111 (CAAAGGGAGGCCAGGAAATTT), polR1 (TCTCTTCTGTTAATGGCCATTGTTTAA), RT1A (GTTGACTCAGCTTGGTTGTAC), R3271 (ACTGTCCATTTGTCAGGATG).

Sequencing conditions: 1 cycle at 96 °C for 2 min, 25 cycles at 96 °C for 10 s, 50 °C for 5 s, and 72 °C for 4 min, with final extension at 72 °C for 7 min. The obtained nucleotide sequences were processed using the Sequencing Analysis Software v. 5.2 (Applied Biosystems, USA).

The sequencing reaction in the ViroSeq system was performed according to the manufacturer’s protocol. Then, sequence processing was performed automatically using ViroSeq HIV-1 Genotyping Software v. 3.0 (Applied Biosystems, USA).

Multiple sequence alignments were performed using the ClustalW module integrated into the AliView v.1.27 software package [24]. When problematic regions in the obtained alignments were identified, additional sequence alignments were performed “manually”. The total length of the alignment was at least 919 nucleotides, which covered the entire protease and partial reverse transcriptase (positions 2253–3171 and more (up to 3554), HXB2-numbering GenBank K03455).

### 2.3. Preliminary HIV-1 Subtyping

Preliminary HIV-1 subtyping was performed using the online tools: COMET HIV-1 [25], HIVdbProgram Sequence Analysis, presented on the website of Stanford University [26], and REGA HIV-1 Subtyping Tool (V3) [27], according to the algorithm shown in Figure 1.

When the HIV-1 genetic variant was unambiguously interpreted with identity from 97 to 100% in all three programs, it was considered to be reliably determined. In case of ambiguous results and/or less than 97% identity, the sequences were subjected to additional analysis. First, we searched for closely related sequences using the BLAST tool (https://www.hiv.lanl.gov/content/sequence/BASIC_BLAST/basic_blast.html accessed on 1 March 2023). If several closely related sequences were found with identity from 97 to 100%, the genetic variant of sequences under investigation was defined as identical to those found. The remaining sequences were subjected to additional recombination analysis.

### 2.4. Recombination Analysis

Sequences with an indeterminate genetic variant were subjected to additional recombination analysis using jpHMM [28], RIP (https://www.hiv.lanl.gov/content/sequence/RIP/RIP.html accessed on 1 March 2023), and Recombination Detection Program [29] tools. Reference nucleotide sequences used for RDP4 program analysis are presented in Appendix A. A recombination event detected by RDP4 was considered valid if it was identified by two or more mathematical models.

Graphical representations of genome maps highlighting recombination points were generated either using the jpHMM v.0.1.4 software or manually created based on RDP4 results using the Recombinant HIV-1 Drawing Tool (https://www.hiv.lanl.gov/content/sequence/DRAW_CRF/recom_mapper.html accessed on 1 March 2023).

### 2.5. Phylogenic Analysis

The analysis was performed for the recombinant forms identified in the study. The best-fit model of nucleotide substitutions was selected using jModelTest v. 2.1.7 according to the Akaike information criterion (AIC) [30]. Epidemiological clusters were determined using the maximum-likelihood analysis implemented by the IQ-TREE [31] with 1000 bootstrap replicates and Shimodaira–Hasegawa (SH-aLRT) test. Clusters with an SH-aLRT support > 0.9 were considered reliable. Visualization and graphical processing of the phylogenetic analysis results were carried out using iTOL v.6 software [32].

Reference sequences were sourced from the database of the Los Alamos Laboratory, USA (https://www.hiv.lanl.gov/content/sequence/HIV/mainpage.html accessed on 1 March 2023).

### 2.6. Drug Resistance Mutation Analysis

In this study, we performed drug resistance analysis among HIV-infected patients not on ART, infected with both viruses of “pure” subtypes and its recombinant forms. The analysis utilized Stanford University’s HIVdb database tool and the Calibrated Population Resistance Tool (CPR) [33].

Since we could not estimate the duration of patients’ infection and the possibility of their treatment in the past, and only knew that they were not receiving ART at the time of genotype analysis, we decided not to call this pretreatment drug resistance or transmitted drug resistance, and introduced the general name of primary resistance (PrimDR) in this context.

### 2.7. Statistical Analysis

The analysis was carried out using the R programming language (RStudio v.1.3.1093, Inc. Software, Boston, USA), as well as the STATISTICA v. 6.0 (StatSoft, Tulsa, USA).

To estimate the trends in HIV-1 recombinant form prevalence in the Russian Federation, the following conditions were established:HIV-1 nucleotide sequences were sourced from a minimum of 4 (50%) federal districts annually;In the event that Condition No. 1 was not met, the option to aggregate data from multiple years or incorporate supplementary sequences from the international database of the Los Alamos Laboratory (https://www.hiv.lanl.gov/content/sequence/HIV/mainpage.html accessed on 1 March 2023) was permitted, ensuring characteristics aligned with the examined sequences (pol gene; coordinates: 2253–3554, HXB2-numbering; with a minimum sequence length of 919 bp).

Categorical data evaluated in the study were presented as proportions and frequencies and their comparison was carried out using the chi-square test (χ2); if unstable, Yates’ correction χ2 or Fisher’s two-sided exact test was used. Estimates of the PrimDR and DRMs prevalence were calculated with 95% confidence intervals (CIs). Differences were considered significant at *p*-value < 0.05. Data visualization was performed using the R programming language.

## 3. Results

### 3.1. Characteristics of the Study Population

The genotyping results of 3178 HIV-infected patients from seven Russian federal districts (FDs) were examined. The majority of patients hailed from the Central FD (1247, 39.2%), followed by the Southern (555, 17.5%), Siberian (460, 14.5%), Volga (251, 7.9%), Far Eastern (243, 7.7%), and Ural (216, 6.8%) FDs. The smallest patient cohort hailed from the Northwestern FD—totaling 206 (6.5%). A male predominance (1775, 55.9%) was noted in the gender distribution of HIV-infected patients (Table 2).

The study population had a median age of 35.0 years, ranging from 24.6 to 45.4 years. The population exhibited an aging trend, primarily comprising individuals of working age, with a range of 23.0 to 48.6 years. The main risk factor was heterosexual contacts (1642, 51.7%), followed by intravenous drug usage (1196, 37.6%), MSM contacts (148, 4.7%), and mother-to-child transmission (113, 3.6%). 

Out of the total patient count, 65 individuals (2.0%) did not have a clearly identified risk factor. Among the cases of HIV-infection, 14 (0.4%) were categorized under the “Other” group. This “Other” group included the following: nosocomial infections, 12 cases; infections acquired during professional medical activities, one case; and a single case resulting from the tattooing process.

### 3.2. Primary HIV-1 Subtyping

The vast majority of the studied PR-RT sequences (*n* = 3178) were classified as sub-subtype A6 (2633, 82.9%), followed by subtype B (227, 7.1%). The combined prevalence of the circulating recombinant forms (CRFs) CRF02_AG and CRF63_02A6 was 4.8% (151 sequences); CRF03_AB (36, 1.1%) was also found. Other genetic variants accounted for less than 1% each and were presented by the following subtypes: A1 (1/3178), C (22/3178), D (1/3178), F1 (1/3178), G (18/3178), and recombinant forms CRF01_AE (1/3178) and CRF11_cpx (1/3178). A difficult-to-interpret genotyping results were obtained for 86 sequences (2.7%), which were preliminarily classified as unique recombinant forms (URFs).

Subsequently, we conducted an analysis of the prevalence of HIV-1 genetic variants among patients categorized into distinct transmission risk groups (Table 3).

Among patients who were infected through heterosexual transmission, a significant prevalence of sub-subtype A6 viruses (1385/1642, 84.3%) was observed. Subtype B dominated (73/148, 49.3%) in the MSM cohort (*p* < 0.001). A high frequency of HIV-1 recombinant forms (CRFs and URFs together) was noted among IDUs (10.2%).

### 3.3. Phylogenetic Analysis of HIV-1 Recombinant Forms

First, we analyzed viruses of recombinant forms first identified directly on the territory of the Russian Federation and widely spread there—CRF03_AB and CRF63_02A6 [14,15].

Sequences attributed to CRF02 and CRF63 forms were included in a unified phylogenetic analysis, owing to the interrelation between these recombinant forms and the relatively comparable structure of their genomes (Figure 2).

The results revealed the following pattern of clustering: 36 sequences were grouped within CRF02 clusters, originating from diverse sources (3 from Africa and 33 from FSU); and 110 sequences were categorized within the CRF63 cluster. In addition, one sequence was clustered separately from the others on the phylogenetic tree and identified as URF.

Furthermore, numerous significant clusters were directly identified among the examined sequences, signifying multiple instances of virus transmission within the country.

Sequences attributed to CRF03_AB formed the significant cluster with the reference sequence first identified in Kaliningrad during the 1980s. This finding further suggests the occurrence of multiple instances of virus transmission within the country (Figure 3).

Two of the identified HIV-1 recombinant forms—CRF01_AE and CRF11_cpx—are rare and uncharacteristic in its genetic structure on the territory of the Russian Federation. We found that CRF01_AE entry into the territory of Russia occurred as a result of its single introduction from the Republic of the Philippines; CRF11_cpx came from Africa (Figure 4).

### 3.4. URFs Found in Russia

The vast majority of URFs were formed by A6/B (70/87, 80.5%), followed by A/G (13/87, 14.9%). A mosaic genome structure was found in 4.6% (4/87) of URFs (Appendix A).

The analysis of the B-segments of AB-unique recombinant genomes revealed the following distribution: 31.4% (22/70) of them contained the B-segment common among FSU countries (B-FSU), 27.1% (19/70) were associated with the B-pandemic clade, 18.6% (13/70) contained B-Caribbean fragments, and 17.1% (12/70) contained B-Thailand fragments [34,35,36,37,38]. Furthermore, four AB-unique recombinant forms (AB-URFs) were identified, characterized by genomes composed of segments from sub-subtype A6 and CRF03_AB.

### 3.5. Prevalence of HIV-1 Recombinant Forms

Sub-subtype A6 was the most common genetic variant circulating among residents from seven Federal Districts. The regions displaying the highest genetic diversity were the Central, Northwestern, Far Eastern, and Southern FDs. (Figure 5).

A notable difference in the HIV-1 recombinant form prevalence was observed across the Federal Districts (FDs), with high rates in the Northwestern (18.5%) and Siberian (15.0%) FDs, and low rates in the Volga (1.9%) and Ural (2.8%) FDs. Within the Northwestern FD, a substantial prevalence of CRF03_AB (12.1%) was noted, while in the Siberian FD, CRF63_02A6 was prominent (13.3%).

Furthermore, a significant upward trend in the frequency of HIV-1 recombinant forms was identified over time. In samples collected during 2011–2012 (N = 835), recombinant forms were present in 4.8% of cases. This prevalence escalated to 7.6% in samples from 2015–2016 (N = 741), and surged to 25.4% in samples from 2019–2020 (N = 1203) (*p* < 0.001) (Figure 6).

A visual assessment revealed a sharp increase in the proportion of recombinants in the period 2017–2018, which is most likely due to the inclusion of sequences from the city of Cherepovets, which is characterized by a high prevalence of CRF03_AB HIV-1. Despite this shift in sampling, the general trend towards an increase in the frequency of the detection of recombinant forms of HIV-1 in the Russian Federation has remained over time.

### 3.6. PrimDR and DRMs between HIV-1 “Pure” Subtypes and Recombinant Forms

We investigated the prevalence of PrimDR and individual DRMs in PR-RT sequences derived from HIV-infected patients not on ART (N = 1494) concerning PIs, NRTIs, and NNRTIs. The prevalence of PrimDR in patients infected with HIV-1 recombinant forms (N = 175) was 11.4% (95% CI, 7.5–17.0%), and in patients infected with HIV-1 “pure” viral subtypes (N = 1319) it was 6.9% (95% CI, 5.7–8.4%) (*p* = 0.044). The overall prevalence of PrimDR in patients not on ART with the 2011–2020 sampling years was 7.4% (95% CI, 6.2–8.9%). The most frequent DRMs were K103N—33,3% (37/111), M184V—32.4% (36/111), G190S—17.1% (19/111), and M46I—10.8% (12/111).

The frequencies of certain DRMs were significantly different between HIV-1 recombinant forms and “pure” subtypes; the major NNRTI-associated mutation K103N was more frequently observed in CRF63_02A6 (65.0% (95% CI, 43.3–81.9%) vs. 26.4% (95% CI, 18.4–36.3%), *p* = 0.002), and the NRTI mutation M184V was more frequently observed in sub-subtype A6 (37.4% (95% CI, 28.1–47.6%) vs. 10.0% (95% CI, 2.8–30.1%), *p* = 0.018) (Figure 7).

The greatest diversity of DRMs was observed in viruses with a unique genome structure that carried mutations to ARV of different classes: G73S to PIs; M41L, D67N, K70R, and M184V to NRTIs; and K103N, Y181C, and G190S to NNRTIs. Multidrug resistance (MDR) was found in 10% (95% CI, 2.8–30.1%) of patients not on ART infected with HIV-1 recombinant forms.

## 4. Discussion

In this study, we analyzed 3178 HIV-1 *pol* sequences obtained from HIV-infected patients from seven Russian federal districts. Our aim was to identify and analyze recombinant forms of HIV-1 in Russia in the last decade of the epidemic using phylogenetic and recombination approaches. The present study showed a wide genetic diversity of HIV-1 with an increasing frequency of recombinant forms in Russia. Phylogenetic analysis revealed both single cases of the introduction of recombinant forms into Russia and their multiple transmission within the country. Different DRMs have been observed between “pure” subtypes and recombinant forms of HIV-1. Our study contributes to understanding the main trends of the HIV epidemic in Russia, assessing its biological and social driving forces, as well as to understanding the dynamics of changes in the HIV-1 genetic structure and evaluating the effectiveness of currently used ART. All nucleotide sequences obtained in the study were deposited in the international database of the Los Alamos laboratory (Appendix A).

This study highlights significant trends in the progression of the HIV epidemic in Russia. Notably, there has been an evident shift towards an “aging” HIV-infected population, along with a notable rise in female involvement compared to the preceding decade. This has resulted in an increase in heterosexual transmission [39]. There is also an increase in the proportion of MSM, but this is most likely due to improved epidemiological investigations and greater “openness” of this cohort to physicians. The median age within the study population was 35.0 years, indicating this ageing trend. Notably, individuals within the active working age bracket (23.0 to 48.6 years) comprised the majority. The male-to-female ratio was 1.3:1. In terms of risk factors, the majority of patients sampled between 2011 and 2020 reported heterosexual contact as the primary mode of transmission (51.7%), followed by intravenous drug use (IDUs) at 37.6%, and men who have sex with men (MSM) at 4.7%.

Our findings revealed that sub-subtype A6, historically widespread in FSU countries, remains the dominant HIV-1 genetic variant in seven FDs (82.6%), followed by subtype B (7.1%). This also reflects the general patterns of the HIV epidemic in Russia, described in previous studies [11]. Obviously, the co-circulation of HIV-1 belonging to different genetic variants, along with high population mobility within a single territory, may contribute to their further recombination and the formation of new genetic variants, as evidenced by studies of HIV-1 recombinant forms detected in Russia—CRF03_AB and CRF63_02A6—and in neighboring countries—CRF02_AG [14,15,40]. According to the results of our study, the proportion of each of the recombinant forms was 1.1% for CRF02_AG and CRF03_AB, and 3.6% for CRF63_02A6. In addition to HIV-1 CRFs, 87 URFs were found, which accounted for 2.7% of all nucleotide sequences examined. Other genetic variants were found with a frequency of less than 2%.

These findings indicate the extensive genetic diversity of HIV-1 in Russia, emphasizing the necessity for ongoing and systematic monitoring. The highest genetic diversity was observed in the Central, Northwestern, Far Eastern, and Southern FDs.

It should be noted that a high frequency of HIV-1 recombinant forms (CRFs and URFs together) was noted among IDUs (10.2%). This is possibly explained by the highest probability of double- and superinfections when using non-sterile instruments (needles/syringes). A number of studies have noted the importance of dual HIV infection for the formation of HIV-1 recombinant forms and the development of drug resistance, even among naїve patients [41,42].

An assessment of the prevalence of HIV-1 recombinant forms in the last decade of the epidemic in Russia revealed a significant increase in their frequency over time (*p* < 0.001). A high prevalence of HIV-1 recombinant forms was observed in the Northwestern (18.5%) and the Siberian (15.0%) Federal Districts, which are the territorial centers of CRF03_AB and CRF63_02A6 origination in Russia, respectively [14,15].

A low prevalence of HIV-1 recombinant forms was observed in the Volga (1.9%) and Ural (2.8%) FDs. This may be associated with the internal migration processes in Russia. For example, in the Volga Federal District (FD), a significant emigration to other regions was observed, coupled with a minimal influx of the population [43].

The phylogenetic analysis of HIV-1 recombinant forms in Russia revealed that CRF01_AE and CRF11_cpx originated from the Republic of the Philippines and Africa, respectively. In the case of CRF03_AB and CRF63_02A6 recombinant viruses, multiple transmissions within Russia were observed, likely due to their widespread presence in the region [14,15]. The most robust phylogenetic clusters encompassed several nucleotide sequences obtained from IDUs.

The overwhelming majority of URFs were composed of A6 and B (80.5%), which are the prevailing genetic variants in Russia [11]. A further 14.9% of URFs were formed by A6 and G, while the remaining 4.6% exhibited a mosaic genome structure. This diversity of URFs indicates an active recombination process within the country, potentially leading to the rapid emergence and widespread dissemination of new CRFs.

An analysis of the B-segments of the AB-URFs genome showed that 31.4% of them contained the B-FSU segment, which was first identified and is still widespread in Ukraine (Mykolaiv) among IDUs [44], and 27.1% of them contained the B-segment related to the pandemic clade, which originated in the USA and later entered Western Europe countries through local epidemics among MSM or through IDUs [34,35,36,37,38]. The distribution of this subtype B is most likely due to the relative geographic proximity of Russia to European countries and the intensification of contacts between them in the late 1990s. The remaining 18.6% of the AB-URFs genome contained B-Caribbean and 17.1% contained B-Thailand fragments distributed in the Caribbean and Thailand, respectively. Tourism might have played a role in their introduction to Russia; however, factors such as geographical remoteness (particularly for the Caribbean region) and the comparatively recent surge in popularity of tourist destinations like Thailand could have led to a slower dissemination, unlike the faster spread seen with B-FSU and pandemic B strains.

Our study revealed statistically significant differences in the prevalence of PrimDR between viruses of “pure” subtypes and recombinant forms (*p* = 0.044). A higher prevalence of PrimDR was observed for recombinants forms (11.4%), which is 1.4% higher than the 10% threshold value [45]. A previous study by other researchers demonstrated the predominance of HIV-1 recombinant forms (96.2%) in a cohort of HIV-infected people in Wuhan (China) where transmitted DR among ART-naїve patients was 23.9% [46]. Also, the prevalence of drug resistance mutations (DRMs) among HIV-1 unique recombinant forms in people under long-term antiretroviral treatment failure from the Yunnan Province (China) was 56.9% [47].

The overall prevalence of PrimDR to any drug classes in the 2011–2020 sampling years was 7.4%, mainly due to NNRTI. Thus, a higher prevalence (*p* = 0.002) of K103N in HIV-1 recombinant forms, in particular, in CRF63_02A6, was observed. The K103N mutation is located in the region of the CRF63_02A6 genome, which includes a fragment belonging to another recombinant form, CRF02_AG. The presence of K103N is primary conditioned by the use of EFV as part of the preferred first-line therapy regimen in Russia [48]. In previous studies, the predominance of K103N among naїve patients infected with CRF63_02A6 and CRF02_AG was noted [49,50].

It is also worth noting that the K103N mutation is not associated with a major impact on viral fitness, and the persistence of a drug-resistant virus without reduced replication capacities is likely after ART interruption [51,52,53]. This suggests that this mutation is beneficial for HIV-1 and is most frequently selected during recombination.

A heightened prevalence of the M184V mutation was noted within sub-subtype A6 (*p* = 0,018). This can be explained by the wide distribution of this HIV-1 genetic variant and the use of 3TC and FTC as the main components of the preferred first-line therapy regimens in Russia. Earlier investigations also demonstrated the predominance of the M184V mutation in subtype A viruses. However, this observation was primarily compared with subtype D viruses in those studies [54].

Quite a high proportion of DR was detected to NNRTIs, which may indicate the need to abandon them in favor of promising drugs of the integrase inhibitor class. Thus, in December 2020, the Ministry of Health of the Russian Federation approved the clinical recommendations “HIV infection in adults”, in which DTG was included in one of the three options of preferred first-line ART regimens for adults along with EFV and ESV. Also, in 2021, the first long-term contract for this drug was conducted [55,56].

Of particular interest at present is also the long-acting injectable treatment for HIV-1, which is being actively introduced in Europe. Multivariable baseline factor analysis across the latter three clinical trials (FLAIR, ATLAS, and ATLAS-2M) recently showed that HIV-1 subtypes A6/A1, characterised by the L74I integrase (IN) polymorphism, were one of the factors associated with an increased risk of virological failure (VF) using dual therapy CAB + RPV (with odds ratio 6.59). These findings emphasize the potential importance of genotyping across the integrase domain [57,58].

The study acknowledges certain limitations, notably the potential for biased assessments due to uneven sample distribution across Federal Districts, with the Central FD contributing the largest portion (39.2%), and the Northwestern FD the smallest (6.5%). Additionally, our findings are confined to the HIV-1 pol region; employing complete genomes may yield more precise insights into the prevalence of recombinant virus forms. Therefore, caution is advised in interpreting the results.

Despite these constraints, we posit that our analysis, encompassing 3178 sequences from HIV-infected patients both receiving ART and not on ART, serves as a crucial component of the molecular genetic monitoring of the HIV infection from 2011 to 2020. This approach is deemed optimal for comprehending shifts in the genetic structure of HIV-1, including its recombinant forms, over time, especially in the absence of universal genotyping and genotypic resistance testing for all HIV-infected individuals prior to therapy.

The findings from our study suggest a rise in the proportion of recombinant forms within the genetic structure of the HIV-infection epidemic, encompassing those with primary resistance. This trend coincides with the spread of the HIV infection beyond vulnerable groups and into the general population. These developments could potentially increase the prevalence of drug resistance prior to treatment, with the potential to exceed the 10% threshold recommended for genotypic resistance testing before initiating therapy in the upcoming years.

## Figures and Tables

**Figure 1 viruses-15-02312-f001:**
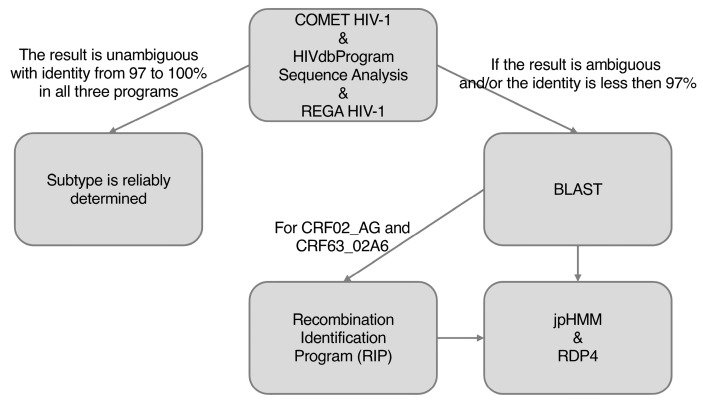
Algorithm for determining HIV-1 genetic variants.

**Figure 2 viruses-15-02312-f002:**
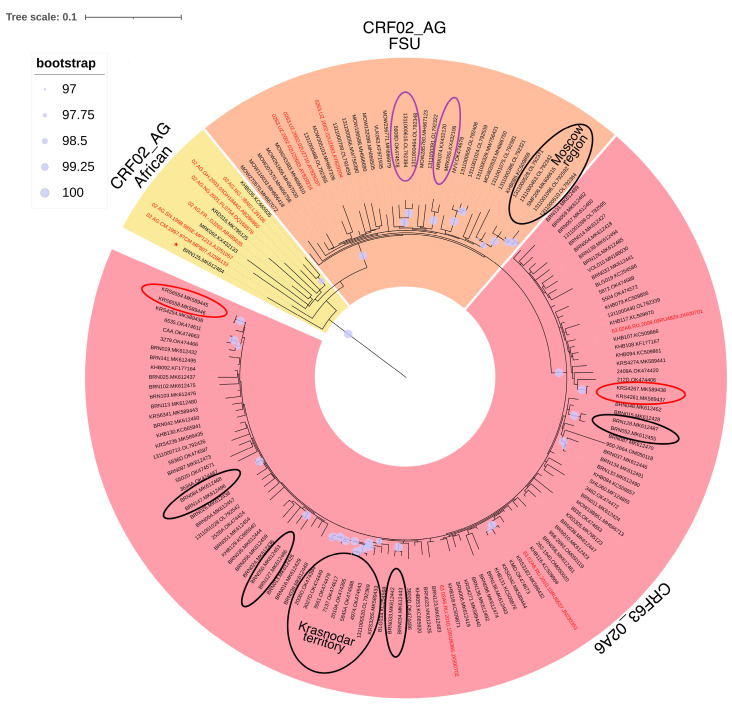
Results of phylogenetic analysis of HIV-1 CRF02_AG and CRF63_02A6 circulating in Russia. Alignment length—1098 bp; nucleotide substitution model—TIM + I + G4. The sequences investigated are marked in black, the reference sequences are marked in red. The yellow cluster contains CRF02 sequences of African origin. The orange cluster includes CRF02 sequences identified in the former Soviet Union (Uzbekistan). The pink colored cluster encompasses sequences attributed to CRF63_02A6. Black ovals outline transmission clusters among IDUs, the largest of which include IDUs from the Moscow region and Krasnodar territory. Purple ovals outline transmission clusters among MSM. Red ovals outline clusters of transmission of infections acquired during professional medical activity. A red asterisk indicates a sequence classified as a URF by the analysis.

**Figure 3 viruses-15-02312-f003:**
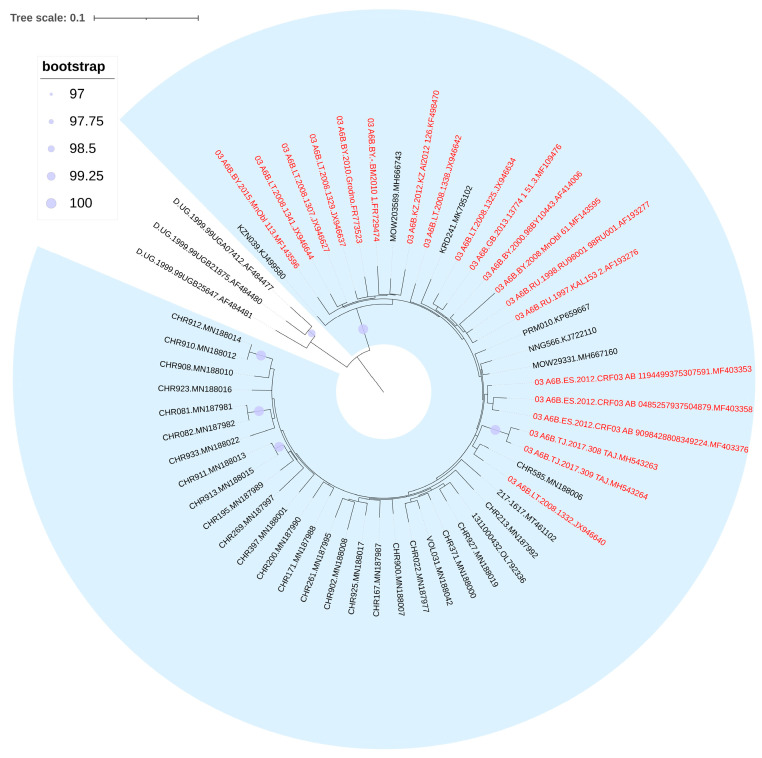
Results of phylogenetic analysis of HIV-1 CRF03_AB circulating in Russia. Alignment length—1098 bp; nucleotide substitution model—TVM + I + G4. The sequences investigated are marked in black, the reference sequences are marked in red.

**Figure 4 viruses-15-02312-f004:**
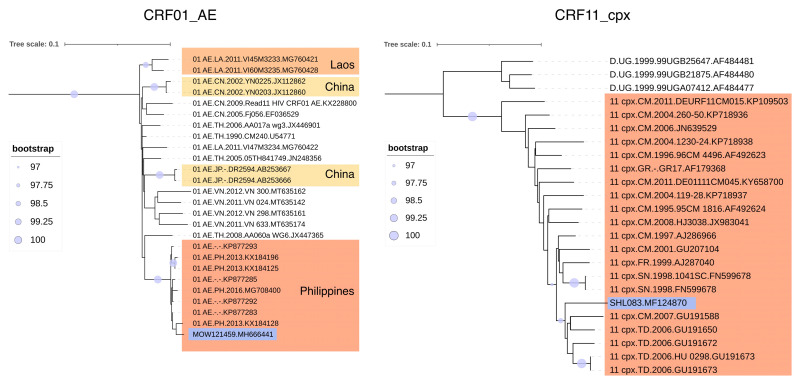
Results of phylogenetic analysis of HIV-1 recombinant forms CRF01_AE and CRF11_cpx identified in Russia. For CRF01_AE: alignment length—1305 bp; nucleotide substitution model—TVM + I + G4; for CRF11_cpx: 1305 bp, GTR + I + G4. The investigated nucleotide sequences are highlighted in purple color.

**Figure 5 viruses-15-02312-f005:**
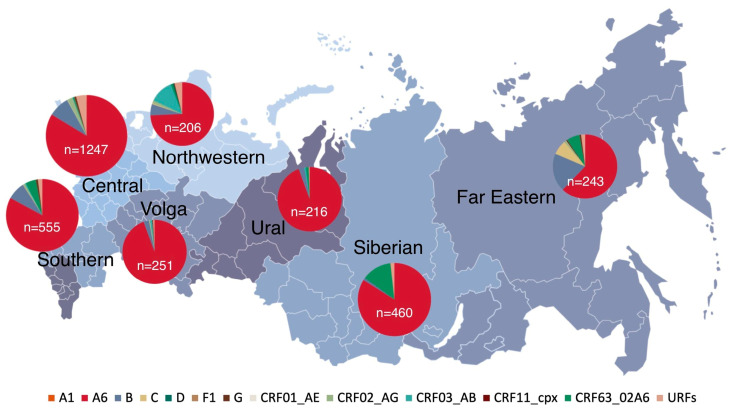
Distribution of HIV-1 genetic variants (by *pol* gene) in the FDs of the Russian Federation in the period 2011–2020 (N_total_ = 3178).

**Figure 6 viruses-15-02312-f006:**
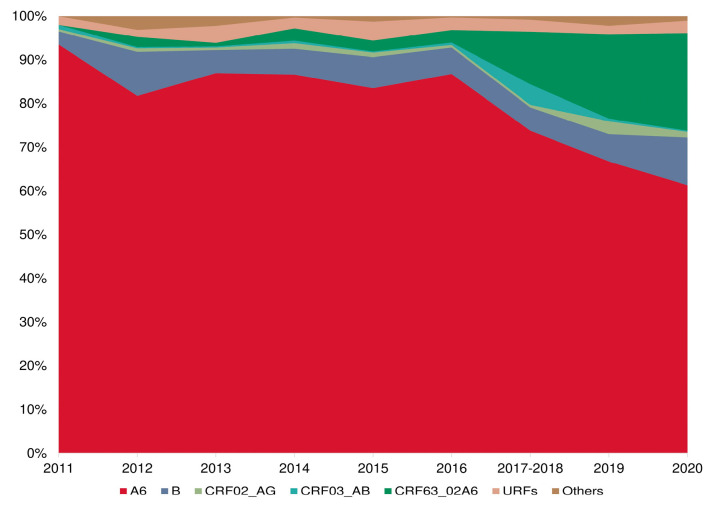
The frequency of HIV-1 recombinant forms and “pure” subtypes in Russia in the period 2011–2020.

**Figure 7 viruses-15-02312-f007:**
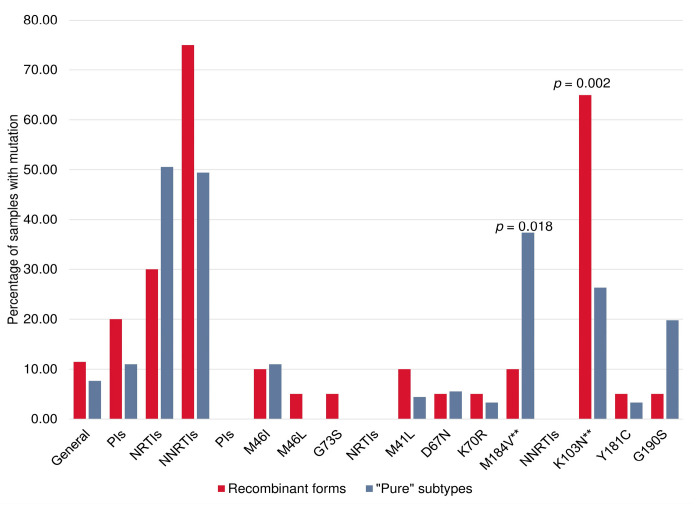
Frequency of DRMs among HIV-1 recombinant forms and “pure” subtypes. Two asterisks indicate mutations for which statistically significant differences were found.

**Table 1 viruses-15-02312-t001:** Amplification protocols and primers used in the study.

Amplification Protocols	Primers		Nucleotide Sequence of Primers (5′→3′)
A (single PR-RT fragment)	1 round	RP1S RP1A	GAAAAAGGGCTGTTGGAAATGTGGAA AAATTTAGGAGTCTTTCCCCATATTACTATGC
	2 round	PROS2 RTOA	GCTAATTTTTTAGGGAAGATCTGGCCTT TGCCTCTGTTAATTGTTTTACATCATTAGTGTG
B (separate fragments encoding PR and part of RT)	1 round	POM (PR) R2726 (PR) F2491 (RT) RT2A (RT)	CCCTAGGAAAAAGGGCTGTTGGA TGGAGTATTGTATGGATTTTCAGGC CCCCTGTCAACATAATTGGA TCTGTATATCATTGACAGTCCAGC
	2 round	F2111(PR) polR1 (PR) RT1A (RT) R3271 (RT)	CAAAGGGAGGCCAGGAAATTT TCTCTTCTGTTAATGGCCATTGTTTAA GTTGACTCAGCTTGGTTGTAC ACTGTCCATTTGTCAGGATG

**Table 2 viruses-15-02312-t002:** Epidemiological characteristics of HIV-infected patients included in the study.

Year	Sex		Median Age	Risk Factor for HIV-1 Infection						Total
	Male, N (%)	Female, N (%)		Heterosexual, N (%)	IDU, N (%)	MSM, N (%)	MTCT, N (%)	Unknown, N (%)	Other, N (%)	
2011	107 (53.5)	93 (46.5)	32.0 [23.6–40.4]	88 (44.0)	99 (49.5)	5 (2.5)	3 (1.5)	4 (2.0)	1 (0.5)	200
2012	309 (48.7)	326 (51.3)	33.0 [23.3–42.7]	271 (42.7)	306 (48.2)	9 (1.4)	34 (5.3)	14 (2.2)	1 (0.2)	635
2013	183 (50.8)	177 (49.2)	34.0 [23.7–44.3]	189 (52.5)	134 (37.2)	10 (2.8)	10 (2.8)	15 (4.2)	2 (0.5)	360
2014	177 (55.1)	144 (44.9)	35.0 [24.3–45.7]	156 (48.6)	122 (38.0)	15 (4.7)	16 (5.0)	9 (2.8)	3 (0.9)	321
2015	237 (60.2)	157 (39.8)	35.0 [24.2–45.8]	200 (50.8)	147 (37.3)	18 (4.6)	22 (5.6)	6 (1.5)	1 (0.2)	394
2016	206 (59.4)	141 (40.6)	36. [25.3–46.7]	198 (57.1)	112 (32.3)	10 (2.9)	20 (5.8)	4 (1.1)	3 (0.9)	347
2017	235 (58.8)	165 (41.2)	36.0 [26.2–45.8]	223 (55.8)	147 (36.8)	18 (4.5)	6 (1.5)	5 (1.2)	1 (0.2)	400
2018	60 (56.1)	47 (43.9)	34.0 [23.0–45.0]	60 (56.1)	29 (27.1)	13 (12.2)	1 (0.9)	3 (2.8)	1 (0.9)	107
2019	164 (62.4)	99 (37.6)	37.0 [26.2–47.8]	168 (63.9)	64 (24.3)	28 (10.6)	1 (0.4)	1 (0.4)	1 (0.4)	263
2020	97 (64.2)	54 (35.8)	39.0 [29.4–48.6]	89 (58.9)	36 (23.8)	22 (14.6)	-	4 (2.7)	-	151
Total	1775 (55.9)	1403 (44.1)	35.0 [24.6–45.4]	1642 (51.7)	1196 (37.6)	148 (4.7)	113 (3.6)	65 (2.0)	14 (0.4)	3178

Abbreviations: IDU, intravenous drug users; MSM, men having sex with men; MTCT, mother-to-child transmission.

**Table 3 viruses-15-02312-t003:** Prevalence of HIV-1 genetic variants among the transmission risk groups.

Genetic Variant	Transmission Risk Groups					
	Heterosexual, N (%)	IDU, N (%)	MSM, N (%)	MTCT, N (%)	Other, N (%)	Unknown, N (%)
A1	1 (0.1)	0	0	0	0	0
A6	1385 (84.3)	1010 (84.4)	62 (41.9)	104 (92.0)	10 (71.4)	62 (95.5)
B	97 (5.9)	55 (4.6)	73 (49.3)	1 (0.9)	0	1 (1.5)
C	13 (0.8)	8 (0.7)	0	0	0	1 (1.5)
CRF01_AE	1 (0.1)	0	0	0	0	0
CRF02_AG/ CRF63_02A6	65 (3.9)	77 (6.4)	3 (2.0)	3 (2.7)	3 (21.4)	0
CRF03_AB	24 (1.5)	12 (1,0)	0	0	0	0
CRF11_cpx	1 (0.1)	0	0	0	0	0
D	1 (0.1)	0	0	0	0	0
F1	0	0	0	1 (0.9)	0	0
G	12 (0.7)	1 (0,1)	5 (3.4)	0	0	0
URFs	42 (2.5)	33 (2.8)	5 (3.4)	4 (3.5)	1 (7.2)	1 (1.5)
Total	1642	1196	148	113	14	65

Abbreviations: IDU, intravenous drug users; MSM, men having sex with men; MTCT, mother-to-child transmission.

## Data Availability

All nucleotide sequences obtained in the study were deposited in the international database of the Los Alamos laboratory.

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
