# Peer review of "Recombinant Forms of HIV-1 in the Last Decade of the Epidemic in the Russian Federation"

_viruses, 2023, doi:10.3390/v15122312_

Round 1

Reviewer 1 Report

Comments and Suggestions for Authors

In the manuscript “Recombinant Forms of HIV-1 in the Last Decade of the Epidemic in the Russian Federation”, the authors described molecular epidemiology study on HIV-1 subtype, recombination, and drug resistance mutations in HIV surveillance sites in Russia from 2011 to 2020. The paper provides useful information regarding the HIV-1 epidemic in Russia, and reveals the trend of HIV transmission across the years. I have the following comments and suggestions.

1. Please describe information regarding proper IRB approval

2. The author provided a thorough introduction of HIV-1 subtypes and distribution in Russia in the literature. However, it is a little too long as introduction. For example, in line 80-93, most of the content can be discussed in the discussion section.

3. Table 2. There is a trend that IDU patients were decreasing over the years and MSM were increasing. Is it a real signal or just sampling bias?

4. Line 258-259. what was the p value, MSM vs. rest routes in subtype B distribution.

5. Line 348. M184V is not an accessory mutation.

6. In section 3.6, the authors called their patients “treatment-naïve”. However, in the methods section, the authors claimed that they did not know the treatment history of the patients, they were not necessarily treatment naïve. In line 477-467, the authors then claimed the sequences were from both pre- and during treatment. Please clarify.

7. Line 375, should be “aging”.

8. The authors can discuss the treatment options and how the resistance mutations would impact the ART options in Russia.

Comments on the Quality of English Language

There are a few Russian alphabet left in the manuscript. For example, Ð¸ in line 344.

Reviewer 2 Report

Comments and Suggestions for Authors

The HIV epidemic circulating in countries of the Russia Federation, including Former Soviet Bloc countries, is the fastest growing epidemic in the world. This epidemic is circulating in the Ukraine, with 8 million persons migrating since the war. It remains important to characterize HIV-1 subtypes and recombinant forms that may impact on viral transmissibility and patterns of acquired drug resistance.     

The study is topical and important. I suggest minor revisions. 

 1. Line 438 paragraph on drug resistance. New findings.  The 5.6 fold higher rates of virological failure for subtype A6 to cabotegravir in the ATLAS and FLAIR trials should be added to the discussion. This emphasizes the potential  importance of genotyping across integrase domain. 

2. Line 449 paragraph. The higher frequency of K103N in subtype D than subtype A may conversely reflect the poorer baseline replicative activity in the African subtype A variant rather than an adverse impact of K103N.    
